# NQO1 is Required for β-Lapachone-Mediated Downregulation of Breast-Cancer Stem-Cell Activity

**DOI:** 10.3390/ijms19123813

**Published:** 2018-11-30

**Authors:** Dong Wook Kim, Je-Yoel Cho

**Affiliations:** Department of Biochemistry, BK21 PLUS Program for Creative Veterinary Science Research and Research Institute for Veterinary Science, College of Veterinary Medicine, Seoul National University, Seoul 08826, Korea; bellocan@snu.ac.kr

**Keywords:** β-lapachone, breast-cancer stem-cell, DLGAP5, mammosphere, NQO1

## Abstract

Cancer stem cells (CSCs) exhibit self-renewal activity and give rise to other cell types in tumors. Due to the infinite proliferative potential of CSCs, drugs targeting these cells are necessary to completely inhibit cancer development. The β-lapachone (bL) compound is widely used to treat cancer development; however, its effect on cancer stem cells remain elusive. Thus, we investigated the effect of bL on mammosphere formation using breast-cancer stem-cell (BCSC) marker-positive cells, MDA-MB-231. MDA-MB-231 cells, which are negative for reduced nicotinamide adenine dinucleotide phosphate (NAD(P)H):quinone oxidoreductase (NQO1) expression, were constructed to stably express NQO1 (NQO1 stable cells). The effect of bL on these cells was evaluated by wound healing and Transwell cell-culture chambers, ALDEFLUOR assay, and mammosphere formation assay. Here, we show that bL inhibited the proliferative ability of mammospheres derived from BCSC marker-positive cells, MDA-MB-231, in an NQO1-dependent manner. The bL treatment efficiently downregulated the expression level of BCSC markers cluster of differentiation 44 (CD44), aldehyde dehydrogenase 1 family member A1 (ALDH1A1), and discs large (DLG)-associated protein 5 (DLGAP5) that was recently identified as a stem-cell proliferation marker in both cultured cells and mammosphered cells. Moreover, bL efficiently downregulated cell proliferation and migration activities. These results strongly suggest that bL could be a therapeutic agent for targeting breast-cancer stem-cells with proper NQO1 expression.

## 1. Introduction

Cancer stem cells (CSCs) have self-renewal activity; however, their population in tumors is very low, ranging from 0.1 to only a few percent. Recent studies identified that CSCs could divide in an asymmetric or symmetric manner depending on the size of the niche [1]. Many studies advanced the understanding of the biology of breast-cancer stem-cell (BCSC) development. Stem cells (SCs) in breast tissue also exhibit self-renewal activity and divide into transient amplifying progenitor cells, which become differentiated cells. Of note, BCSCs may originate from stem cells, transient amplifying progenitor cells, or differentiated cells through genetic alteration by environmental stimuli, indicating that the development of BCSCs is a very dynamic process [2]. The population of BCSCs in tumors is very small and expresses stem-cell-associated markers. Well-defined BCSC markers for the identification of BCSCs in tumor tissue and cell lines are cluster of differentiation 44 positive (CD44^+^)/CD24^−^/aldehyde dehydrogenase positive (ALDH^+^) phenotypes, although other possible human BCSC markers, such as CD49f^+^, were proposed [2,3,4,5,6,7]. A characteristic of BCSCs is an enhanced resistance to drugs, radiation, and cell stress that is strongly associated with metastasis and relapse. Four major therapy resistance mechanisms in BCSCs were proposed. Firstly, BCSCs have an active ATP-binding cassette (ABC) transporter system that pumps out anthracycline or taxanes, which are two key drugs for breast-cancer treatment [8,9]. Secondly, aldehyde dehydrogenase (ALDH1) activity is highly increased [10]. ALDH1 facilitates oxidation of intracellular aldehydes to carboxylic acids and retinoic acids, as well as gamma-amino butyric acid (GABA) biosynthesis. Moreover, ALDH1 induces radioresistance in BCSCs via direct removal of oxygen radicals and indirect production of the antioxidant compound nicotinamide adenine dinucleotide (NAD). Thirdly, BCSCs have an active DNA double-strand break (DSB) repair system (induced particularly by ionizing radiation), which repairs through two distinct and complementary systems of homologous recombination (HR) and non-homologous end-joining (NHEJ) [11]. Finally, BCSCs have specific mechanisms to protect themselves from the genotoxic effects of reactive oxygen species (ROS) by overexpressing ROS-scavenging genes, such as superoxide dismutase, catalase, and glutathione peroxides [12,13].

The β-lapachone (bL) compound is an *o*-naphthoquinone extracted from plants, and has many pharmacological effects [14]. In the last decade, many studies to elucidate the effect of bL on cancer development were conducted and showed that bL has an inhibitory effect on various cancers such as epidermoid laryngeal cancer [15], as well as prostate [16,17,18,19], colon [20,21], ovarian [19], lung [22], and breast-cancer [23]. Additionally, bL was identified as suppressing cancer proliferation by directly interacting with and inhibiting the catalytic activity of DNA topoisomerase I [24], by inducing apoptosis or necrosis through the release of mitochondrial cytochrome C from mitochondria [25] and poly(ADP-ribose) polymerase (PARP) cleavage [26,27], by blocking the lethal DNA damage repair (PLDR) system [28], by inducing gap 1/synthesis (G1/S) cell-cycle arrest [29], and by activating c-Jun NH_2_-terminal kinase [30] and caspases [31]. Recent data revealed that reduced NAD phosphate (NAD(P)H):quinone oxidoreductase (NQO1) is a critical enzyme in bL-mediated inhibition of cancer proliferation [23,32]. NQO1 converts cellular NAD(P)H into NAD(P)^+^, and bL accelerates this reaction by accepting hydrogen released from NAD(P)H, thereby increasing the cellular level of NAD(P)^+^ [33]. Moreover, reduction of bL by NQO1 leads to futile cycling, resulting in the generation of superoxide and hydrogen peroxide [34,35,36]. In addition, direct reduction of bL by NADPH cytochrome p450 reductase (P450R) and NADH cytochrome b5 reductase (b5R) can generate superoxide and hydrogen peroxide [37,38].

NQO1 is a multifunctional antioxidant enzyme, and its expression and deletion are closely related with decreased and increased susceptibilities to oxidative stress, respectively [39]. The expression level of NQO1 is highly increased in various cancers, including prostate cancer [38], hepatocellular carcinoma [40], and breast-cancer [41,42], indicating an essential role of NQO1 in cancer development. Oxidative stress is one of the most important regulatory mechanisms for cancer stem cells. Oxidative stress in cancer cells is known to play a key role in either the initiation and progression of cancer or in the induction of cancer cell death, depending on the intensity of the oxidative stress [43,44]. CSCs and normal stem cells have a low level of ROS, which is an important factor for stem-cell maintenance in a stressful environment, meaning that there is high expression of ROS-scavenging molecules [45]. Recent data suggest that nuclear factor erythroid 2-related factor 2 (NRF2), which is a well-known NQO1 upstream regulator, is involved in the maintenance of quiescence, survival, and stress resistance of CSCs [46].

In this study, we focused on evaluating the effect of bL on the activity of BCSCs expressing NQO1, to see whether bL could be used to treat BCSCs expressing NQO1. We found that bL induces cell death and disruption of mammospheres derived from MDA-MB-231 cells (BCSC-positive marker cells) in an NQO1-dependent manner. Our data strongly indicated that bL has an inhibitory effect on cell proliferation in an NQO1-dependent manner, suggesting that certain cancer stem cells expressing NQO1 could be potential therapeutic targets of bL.

## 2. Results

### 2.1. The Cellular Expression Level of NQO1 is Negatively Correlated with That of BCSC Markers

At first, we investigated whether NQO1 expression is related to the expression levels of BCSC markers. We compared two breast-cancer cell lines, MCF7 and MDA-MB-231, which are known as negative and positive cell lines for BCSC markers, respectively [47] Quantitative RT-PCR indicated that the messenger RNA (mRNA) expression level of NQO1 is very low in BCSC-marker-positive cells (MDA-MB-231) compared to BCSC-negative cells (MCF-7) (Figure 1A). Two well-known breast-cancer stem-cell markers, CD44 and ALDH1 family member A1 (ALDH1A1), were examined. Interestingly, CD44 showed a high expression level in MDA-MB-231 cells (Figure 1B), but ALDH1A1 did not, although it showed an increased trend (Figure 1C). In addition, we found that the expression of discs large (DLG)-associated protein 5 (DLGAP5), which is known to increase stem-cell proliferation and be highly expressed in cancers [48,49,50,51], was also higher in MDA-MB-231 compared with MCF7 cells (Figure 1D), indicating that DLGAP5 may positively regulate BCSC proliferation. The protein expression levels of the same genes submitted for quantitative RT-PCR were verified by Western blot analysis. As shown in Figure 1E, NQO1 was only detected in MCF7 cells. Moreover, CD44 and DLGAP5 were highly expressed in MDA-MB-231 cells, but ALDH1A1 showed no difference or was slightly higher in MDA-MB-231 cells, consistent with the mRNA expression level. Taken together, these results indicated that NQO1 is related to the expression of BCSC markers.

### 2.2. β-Lapachone-Mediated NQO1 Activation Regulates DLGAP5 and CD44 Expression Levels

To gain insight into the possible mechanism via which NQO1 regulates DLGAP5 and CD44 expression, we created MDA-MB-231 cells stably expressing either NQO1 (NQO1 stable cells) or the vector control (control cells). The expression of each gene was compared in control cells and in two different clones of NQO1 stable cell lines with or without bL. Interestingly, the gene expression of DLGAP5 and CD44 was downregulated by bL treatment in the presence of NQO1 in MDA-MB-231 cells, but not in control cells, indicating that NQO1 is required for the bL-mediated downregulation of these genes (Figure 2A,B). In contrast, the ALDH1A1 expression level was not altered by bL treatment regardless of NQO1 expression in both control and NQO1 stable cell lines (Figure 2C). To verify the effect of bL-mediated NQO1 on protein expression, Western blot analysis was performed after bL treatment on control and NQO1 stable cells (Figure 2D). As expected, bL treatment did not affect the protein expression levels of DLGAP5, CD44, or ALDH1A1 in control cells. Interestingly, the DLGAP5 protein level was increased by NQO1 expression alone, but bL treatment dramatically decreased the DLGAP5 protein expression in NQO1 stable cells. Moreover, CD44 expression was not affected by NQO1 expression alone, but was also decreased by bL treatment in NQO1 stable cells. These results imply that DLGAP5 is upregulated by NQO1 alone via an unknown mechanism, and that bL is essential for NQO1-mediated downregulation of both DLGAP5 and CD44 gene and protein expression. Unexpectedly, ALDH1A1 was also downregulated by bL treatment in NQO1 stable cells, which was different from the result shown in the mRNA expression pattern (Figure 2C), suggesting that NQO1 activation by bL might regulate ALDH1A1 expression at the post-translational modification level (Figure 2D).

### 2.3. Sirtuin 1 (SIRT1) Is Not Involved in bL-NQO1-Mediated Gene Expression and Cell Death

SIRT1 is an NAD^+^-dependent deacetylase and regulates gene expression by regulating acetylation on proteins [52]. Because SIRT1 is observed in both the cytosol and nucleus, its localization is regarded as an important event in the regulation of cell proliferation [52]. In addition, NQO1 activated by bL accelerates the conversion of NADH to NAD^+^, and increased cellular NAD^+^ levels may affect cancer cell proliferation. Therefore, we hypothesized that a cellular NAD^+^ level increased by bL-NQO1 may activate SIRT1 and regulate BCSC marker gene expression. To verify our hypothesis, we firstly examined SIRT1’s cellular localization after bL treatment. We fractionated NQO1 stable cells after treatment with bL for 24 h. NQO1 was observed mainly in the cytoplasmic fraction, and DLGAP5 and ALDH1A1 were observed in the nucleus. Notably, the DLGAP5 and ALDH1A1 protein levels were again downregulated by bL treatment in the presence of NQO1 expression (Figure 3A). However, we could not find any difference in SIRT1 protein amount in the cytoplasmic and nuclear fractions by bL treatment. Moreover, treatment of bL for 15 and 30 min did not affect SIRT1 localization (Figure 3B). Finally, we performed tests with an SIRT1 inhibitor to see whether SIRT1 is involved in bL-NQO1-mediated cell survival (Figure 3C). In control cells, there were no differences in cell survival among the bL-only treatment, bL + SIRT1 inhibitor (EX-527), and bL + NQO1 inhibitor (ES 936) groups. As expected, the NQO1 inhibitor, but not the SIRT1 inhibitor, efficiently blocked bL-mediated cell death. These results indicated that SIRT1 is not involved in bL-NQO1-mediated cell death.

### 2.4. NQO1 Activated by bL Decreases Cell Proliferation and Migration

To determine the effect of bL on cell viability, cell proliferation and migration assays were performed with or without bL. For the cell proliferation assay, cell numbers were counted after treatment with bL for 24 h. The bL treatment dramatically inhibited cell proliferation in NQO1 stable cells (Figure 4A). Notably, NQO1 expression alone did not affect cell proliferation, as there was no difference among cell lines without bL treatment. Wound healing (Figure 4B) and migration assays using a Transwell system (Figure 4C) showed that bL treatment significantly decreased the cell migration ability at indicated times. Importantly, NQO1 expression alone did not show any effect on cell migration, which is consistent with the results of the cell proliferation assay. Taken together, these results indicated that bL is necessary for NQO1-mediated inhibition of cell proliferation and migration.

### 2.5. ALDH1 Activity Is Decreased by bL through NQO1

The bL treatment decreased the ALDH1A1 protein expression, but not mRNA expression, in an NQO1-dependent manner (Figure 2C,D). Therefore, we investigated whether NQO1 activated by bL treatment could directly affect the activity of ALDH1. To this end, we took advantage of the ALDEFLUOR assay system. Using a cell-permeable substrate of ALDH1, endogenous ALDH1 activity was evaluated according to the intensity of green fluorescence (Figure 5A). Cells with weak green fluorescence (Aldefluor-low) were slightly increased in control cells upon bL treatment. However, bL significantly increased the number of Aldefluor-low cells in NQO1 stable cells. Quantification analysis identified that bL treatment increased the number of Aldefluor-low cells approximately twofold in control cells and approximately 3.7-fold and 3.4-fold in the two NOQ1 stable cells (Figure 5B). These results indicated that bL could decrease endogenous ALDH1 activity in an NQO1-dependent manner.

### 2.6. Mammosphere Formation Was Inhibited by bL-NQO1

The bL compound efficiently inhibited the proliferation of MDA-MB-231 cells, which are positive for BCSC markers, and decreased the expression and activity of CD44 and ALDH1A1, in an NQO1-dependent manner. In addition, DLGAP5 expression, known to increase stem-cell proliferation, was downregulated by bL treatment. These observations led us to test whether bL could suppress mammosphere formation. Cells were cultured in mammosphere media for one or two weeks, and a round type of stem-cell-like phenotype was observed in two-week culture (Figure 6A). Interestingly, NQO1 expression alone seemed to increase the size of mammospheres compared to control cells. Strikingly, the size of mammospheres formed after culturing for one or two weeks was apparently decreased by bL treatment. Next, we tested whether the BCSC markers that we measured in previous results were also altered. Protein expression levels of BCSC markers were analyzed using protein lysates obtained from the cells used in Figure 6A. CD44, DLGAP5, and ALDH1A1 expression levels were dramatically decreased by bL treatment in NQO1-expressing cells (Figure 6B). Notably, NQO1 alone did not show any effect on these protein expression levels, as also shown in previous results. Taken together, these results suggested that the bL-mediated inhibition of mammosphere formation required NQO1 expression.

## 3. Discussion

In this paper, we reported that bL could decrease BSCS marker expression and inhibit the ability of breast-cancer cells to form mammospheres in an NQO1-dependent manner. Interestingly, NQO1 expression alone did not affect BSCS marker expression, cell proliferation, and migration. This result may indicate that basal activity of NQO1 in BCSC is not sufficient to induce cell death, but rather it could increase mammosphere formation shown in Figure 6A. Therefore, over-activation of NQO1 seems to be essential for NQO1-mediated BCSC death. In addition, we also identified that the expression of DLGAP5, which is known to stimulate stem cell proliferation, was decreased by bL treatment in an NQO1-dependent manner. The expression levels of CD44 and DLGAP5, but not ALDH1A1, are regulated at the transcriptional level. The activity of ALDH1A1 was reduced by bL treatment, as evidenced by the ALDEFLUOR assay, meaning that bL has multiple inhibition mechanisms for cell proliferation and survival.

In an attempt to find a downstream mediator of NQO1, we considered a major metabolite produced by NQO1, specifically, NAD^+^, which regulates gene expression and protein activation by activating the sirtuin family [53]. The sirtuin family was identified as a major target of NAD^+^, and among its members, SIRT1 is widely studied in cancer development [54]. Unlike other sirtuin family members, SIRT1 can shuttle between the nucleus and cytoplasm and was reported to be a tumor suppressor or tumor promoter in various cancers [54]. Furthermore, a recent study identified that SIRT1 could deacetylate and deactivate mixed-lineage leukemia 1 (MLL1), a well-known histone methyltransferase of histone H3 Lys4 (H3K4) [55], which is indicative of SIRT1 involvement in the transcriptional regulation of genes. Therefore, we wanted to see whether SIRT1 is a key mediator in bL-NQO1-induced cell death. Unexpectedly, the cell fractionation experiment and fluorescence microscope analysis showed that SIRT1 localization was not altered by bL treatment, even though the protein expression levels of DLGAP5 and ALDH1A1 were still decreased by bL treatment. These results mean that SIRT1 translocation into the nucleus is not necessary in bL-NQO1-induced cell death, suggesting that there is an alternative cell-death pathway induced by bL-NQO1 in breast-cancer, although many reports suggest that the NAD–SIRT1 axis could be a key suppressor in cancer proliferation and survival. One of the possible mechanisms of bL-NQO1-mediated cell death is via hydrogen peroxide. Indeed, bL could generate superoxide and hydrogen peroxide by virtue of NADPH cytochrome p450 reductase and NADH cytochrome b5 reductase, as well as NQO1 [27,35,36,37,39]. Increased hydrogen peroxide may activate PARP and caspase 3, which was already identified as apoptotic and necrotic in other studies [27,31].

CD44 and ALDH1A1 were used as positive BCSC markers, and their expression levels were downregulated by bL treatment in an NQO1-dependent manner. Interestingly, a decreased expression level of CD44 was observed at both the mRNA and protein levels; therefore, CD44 expression is regulated at the transcriptional level by bL. In contrast, ALDH1A1 expression is only downregulated at the protein expression level, meaning that bL may regulate post-translational modification. In addition, we found that the expression level of DLGAP5, which is known to increase stem-cell proliferation, was dramatically decreased by bL treatment. DLGAP5 was identified to be upregulated in hematopoietic progenitor cells and was proposed to play an important role in the proliferation of stem cells [48]. Meanwhile, other studies also identified that DLGAP5 was detected in various cancers, including hepatocellular carcinoma and ovarian cancer cells [50,56], suggesting its potential role as a fetal oncoprotein. Therefore, bL-mediated downregulation of CD44, ALDH1A1, and DLGAP5 could be an essential mechanism of inhibiting cancer cell proliferation, even though the suppression mechanism by bL is different. 

To determine the effect of bL on BCSC proliferation in vitro, we took advantage of the mammosphere formation assay, which is a suitable method to evaluate stem-cell-like features of breast-cancer cells. Compared with a one-week culture, a two-week culture was sufficient to induce the formation of large mammospheres; subsequently, bL was added to the media. The bL-induced inhibition of mammosphere formation was clearly evident in both one- and two-week-culture cells. Since SIRT1 was not involved in bL-NQO1-mediated cell death in our experiments, another possible pathway of inducing cell death and inhibiting mammosphere formation may be a robust increase in hydrogen peroxide. In our mammosphere formation experiment, we found that MDA-MB-231 cells stably expressing NQO1 formed slightly larger mammospheres than did control cells in the absence of bL. Indeed, several lines of evidence identified that increased NQO1 expression in cancer stem cells may increase stem-cell proliferation and survival by removing excess hydrogen peroxide. Therefore, the key cellular mechanism in bL-induced cell death is to robustly increase the endogenous level of hydrogen peroxide to an extent that cells cannot remove it.

Another possible effect of bL on BCSC proliferation may be related to CSC niches that are specialized microenvironments regulating stem cell fate via cell–cell contact and secreted factors. Various cells within the CSC niche produce many factors such as tumor necrosis factor alpha (TNFα), C–X–C motif chemokine ligand 12 (CXCL12), interleukin 8 (IL-8), IL-6, Gremlin1, transforming growth factor beta (TFGβ), vascular endothelial growth factor A (VEGF-A), matrix metalloproteinase 2 (MMP2), MMP3, MMP9, and MMP10 that stimulate CSC self-renewal, induce angiogenesis, and recruit immune cells [57]. The bL compound may regulate CSC activity by suppressing direct production of these factors or inhibiting their activated signaling pathways. Indeed, bL showed inhibitory activity on TNFα secretion from macrophages induced by lipopolysaccharide (LPS) [58], on secretion of IL-6, IL-1β, and MMPs (MMP3, MMP8, and MMP9) from activated microglia [59], on secretion of VEGF in human multiple myeloma cells [60], and related signaling pathways such as nuclear factor kappa B (NF-κB) activation. Therefore, we could not rule out the possible effect of bL on the activity of BCSCs through its regulation of many factors secreted from cells in CSC niches, which raises the need for elucidation in future studies.

## 4. Materials and Methods

### 4.1. Cell Culture and Stable Cell Line Establishment

Two human breast-cancer cell lines, MCF7 (KCLB No. 30022) and MDA-MB-231 (KCLB No. 30026), were purchased from Korean cell line bank (Seoul, South Korea) and cultured in Dulbecco’s modified Eagle medium (DMEM) with 10% fetal bovine serum (FBS) in a humidified incubator at 37 °C and 5% CO_2_. Cells used in this study were tested for mycoplasma contamination. An NQO1-negative cell line, MDA-MB-231, was transiently transfected with an NQO1 expression vector (pEFIRES-NQO1) or a control vector containing a puromycin (Sigma-Aldrich, St. Louis, MO, USA) selection marker using Lipofectamine 3000 reagent (Thermo Scientific, Bellefonte, PA, USA), according to the manufacturer’s instructions. After 24-h transfection, the cell medium was changed to a medium containing puromycin at a final concentration of 2 µg/mL, renewed every two days. Two independent cell clones with puromycin resistance were used for the experiments. 

### 4.2. Quantitative Real-Time PCR

Total RNA was isolated from MCF7 and MDA-MB-231 cells using an RNeasy kit (QIAGEN, Hilden, Germany) as per the manufacturer’s instructions. Quantitative real-time PCR was performed as previously reported [61]. The primers used were as follows: NQO1 forward, 5′–AAAGGACCCTTCCGGAGTAA–3′; NQO1 reverse, 5′–CCATCCTTCCAGGATTTGAA–3′; DLGAP5 forward, 5′–CCAGTCGACACAGGAAGGAT–3′; DLGAP5 reverse, 5′–CATTGCCCTTGGCTTAACAT–3′; CD44 forward, 5′–TGGACAGGACAGGACCTCTT–3′; CD44 reverse, 5′–AGGTCCTGCTTTCCTTCGTG–3′; ALDH1A1 forward, 5′–TTCGAAGGAGTGTTGAGCGG–3′; ALDH1A1 reverse, 5′–AACACTGTGGGCTGGACAAA–3′; and Glyceraldehyde 3-phosphate dehydrogenase (GAPDH) forward, 5′–TCCAAAATCAAGTGGGGCGA–3′; GAPDH reverse, 5′–ATGACGAACATGGGGGCATC–3′. GAPDH was used as the normalization reference for each sample. 

### 4.3. Western Blot Analysis

Western blot analysis was performed as previously reported [62]. Briefly, whole-cell lysates were prepared in radioimmunoprecipitation assay (RIPA) buffer, and isolated total proteins were subjected to SDS-PAGE and membrane transfer for the incubation of the following primary antibodies (Ab): anti-NQO1 Ab (Abcam, Cambridge, United Kingdom), CD44 (Abcam, Cambridge, United Kingdom), ALDH1A1 (Abcam, Cambridge, United Kingdom), DLGAP5 (Abcam, Cambridge, United Kingdom), β-actin (Sigma-Aldrich, St. Louis, MO, USA), SIRT1 (Santa Cruz, Dallas, TX, USA), Lamin b (Santa Cruz, Dallas, TX, USA), and β-tubulin (AbFrontier, Seoul, South Korea). The membrane was developed using a chemiluminescence detection system (ATTO corporation, Tokyo, Japan) and exposed to an X-ray film.

### 4.4. Immunocytochemistry

NQO1 stable cells were seeded into a six-well plate and treated with bL (Santa Cruz, Dallas, TX, USA) at a final concentration of 2 µM with a vehicle control (dimethyl sulfoxide (DMSO)) for 24 h, or as otherwise indicated. After treatment with bL, the cells were fixed with 4% paraformaldehyde for 10 min at room temperature (RT), washed with phosphate-buffered saline (PBS) three times, and then permeabilized with 0.1% Triton X-100 for 10 min at RT. The cells were incubated with primary SIRT1 antibodies (1:100) for 3 h at RT, followed by incubation with fluorescein-labeled goat anti-mouse immunoglobulin G (IgG) antibody (Thermo Scientific, Bellefonte, PA, USA). Cell images were captured under a new hybrid microscope (Echo, San Diego, CA, USA).

### 4.5. Wound Healing Assay

Control and NQO1 stable cells were seeded into a six-well plate containing the culture medium (DMEM supplemented with 10% FBS). After 24 h, the cells were wounded with a p200 tip, washed with 1× PBS, and then treated with bL (2 µM) and DMSO. Pictures were taken at the 18-h time point after bL treatment.

### 4.6. ALDEFLUOR Assay

An ALDEFLUOR assay was performed using an ALDEFLUOR kit (Stemcell Technologies, Cambridge, MA, USA), according to the manufacturer’s instruction. Briefly, control and NQO1 stable cells were seeded into a six-well plate and cultured for 24 h at 37 °C in an incubator supplied with humidified air containing 5% CO_2_. Twenty-four hours later, bL (2 µM) was added to the culture media, and the cells were cultured for an additional 24 h; next, 10 µL of activated ALDEFLUOR was added to the media for 45 min. The cells were washed with PBS three times, and pictures were taken under a hybrid microscope (Echo, San Diego, CA, USA). 

### 4.7. Migration Assay

Cell migration activity was tested using a 5.0-μm-pore polycarbonate membrane inserted Transwell cell culture chambers (Corning, New York, NY, USA). Cells (6 × 10^4^) of control and NQO1 stable lines were seeded into the upper chamber with or without bL (2 µM). After incubation for 7 h at 37 °C, cells on the lower surface were fixed with 4% paraformaldehyde for 5 min. The cells were washed three times with PBS and stained with 1% crystal violet in 2% ethanol for 20 min; next, the cells were again washed three times with PBS. Pictures were taken under a microscope (Echo, San Diego, CA, USA).

### 4.8. Mammosphere Formation

Mammosphere formation was performed using MammoCult media (Stemcell Technologies, Cambridge, MA, USA) supplemented with 4 µg/mL heparin and 0.48 µg/mL hydrocortisone, according to the manufacturer’s instructions. In brief, 4 × 10^4^ control and NQO1 stable cells were seeded into ultralow adherent six-well plates (Stemcell Technologies, Cambridge, MA, USA)) and incubated with MammoCult media for one or two weeks. Then, bL (2 µM) was added for 24 h, and the cell morphology was observed under a hybrid microscope (Echo, San Diego, CA, USA).

### 4.9. Cytoplasmic and Nuclear Fractionation

NQO1 stable cells were seeded into a 100-mm culture dish, cultured for 24 h, and then treated with bL (2 µM) for an additional 24 h. Cytoplasmic and nuclear fractionation was performed using a NE-PER Nuclear and Cytoplasmic Extraction Kit (Thermo Scientific, Bellefonte, PA, USA), according to the manufacturer’s instruction; β-tubulin and lamin B were used as cytoplasmic and nuclear markers, respectively. 

### 4.10. Statistical Analyses

Data are expressed as means ± standard error of the mean (SEM). Significant differences between groups were calculated using Student’s *t*-test with GraphPad Prism 5 (GraphPad Software, La Jolla, CA, USA). A *p*-value <0.05 was considered as statistically significant.

## 5. Conclusions

In this study, we found that bL suppressed the ability of mammosphere formation of MDA-MB-231 cell by downregulating CD44, DLGAP5, and ALDH1A1 in an NQO1-dependent manner. These results strongly suggest that bL could be a potential therapeutic agent targeting breast-cancer stem cells with proper NQO1 expression.

## Figures and Tables

**Figure 1 ijms-19-03813-f001:**
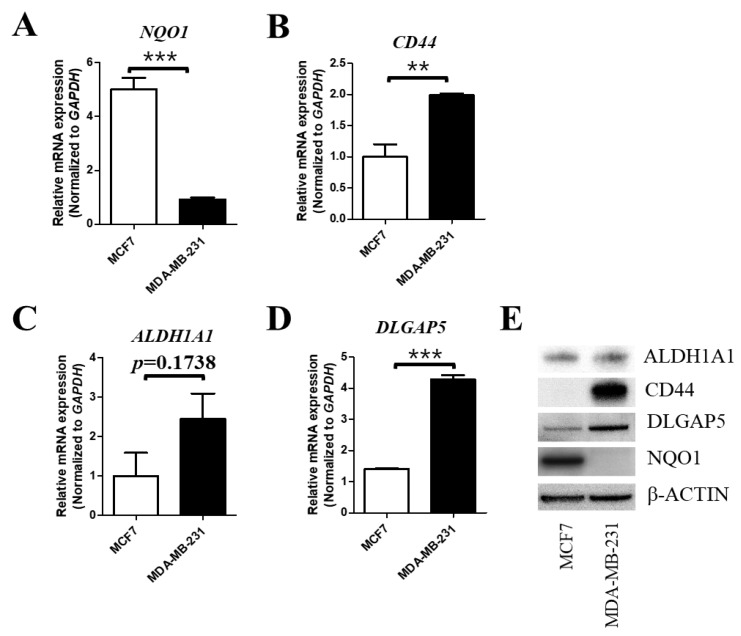
Different expression levels of breast-cancer stem-cell (BCSC) markers between MCF7 and MDA-MB-231 cell lines. (**A**–**D**) The messenger RNA (mRNA) expression levels of reduced nicotinamide adenine dinucleotide phosphate (NAD(P)H):quinone oxidoreductase (NQO1), discs large (DLG)-associated protein 5 (DLGAP5), cluster of differentiation 44 (CD44), and aldehyde dehydrogenase 1 family member A1 (ALDH1A1) were compared using quantitative real-time PCR, as described in Section 4. Glyceraldehyde 3-phosphate dehydrogenase (GAPDH) was used as an internal control, and each expression level was normalized to that of *GAPDH*. The data are presented as means ± standard error of the mean (SEM); *n* = 3; ** *p* < 0.01, *** *p* < 0.001. (**E**) Cell lysates obtained from MCF7 and MDA-MB-231 cells were subjected to Western blot analysis to measure the protein expression level of BCSC markers determined by quantitative RT-PCR; β-actin was used as a loading control.

**Figure 2 ijms-19-03813-f002:**
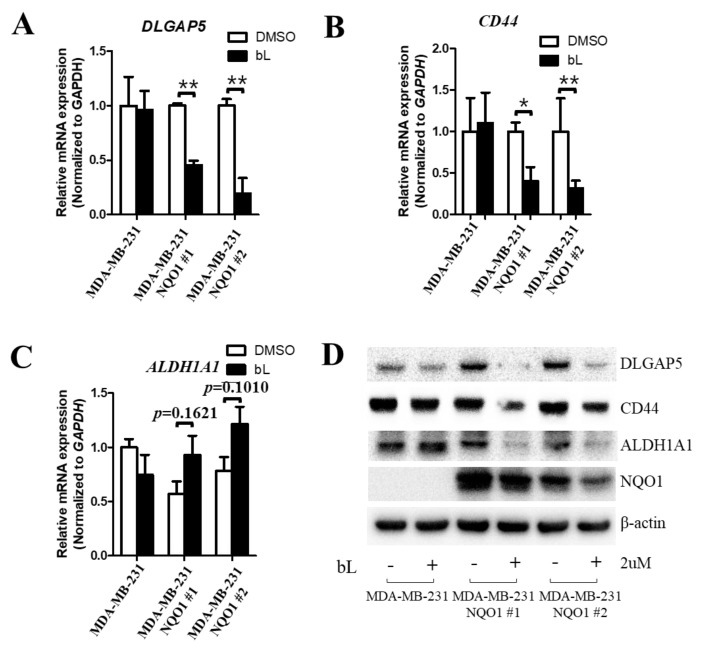
The β-lapachone (bL) compound suppresses the expression of BCSC markers in an NQO1-dependent manner. (**A**–**C**) The mRNA expression levels of DLGAP5, CD44, and ALDH1A1 were compared among MDA-MB-231 and two independent clones of NQO1 stable cells (NQO1 #1 and #2) with or without bL (2 µM) over a 24-h treatment. *GAPDH* was used as an internal control, and each expression level was normalized to that of *GAPDH*. The data are presented as means ± SEM; *n* = 3; * *p* < 0.05, ** *p* < 0.01. (**D**) Protein expression levels of DLGAP, CD44, and ALDH1A1 were compared between MDA-MB-231 and two independent clones of NQO1 stable cells (NQO1 #1 and #2) with or without bL (2 µM) for a 24-h treatment; β-actin was used as a loading control.

**Figure 3 ijms-19-03813-f003:**
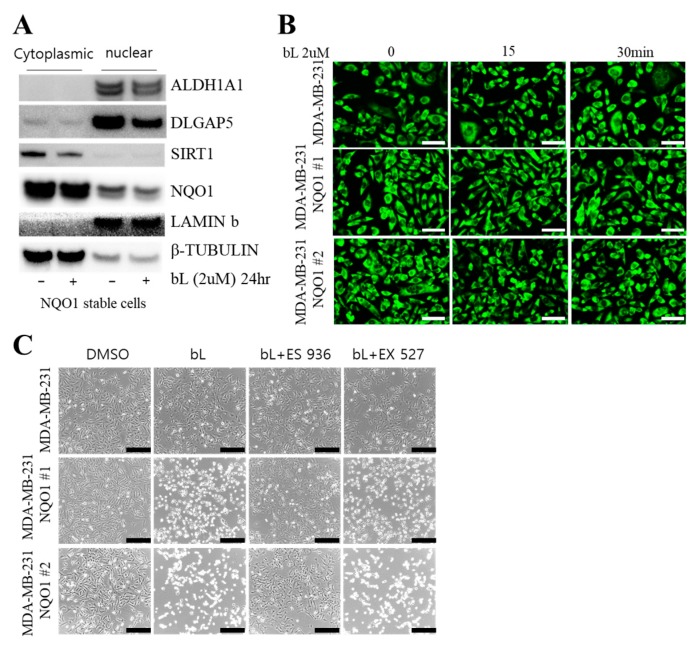
Sirtuin 1 (SIRT1) is not involved in bL-NQO1-mediated cell death. (**A**) Cytoplasmic and nuclear fractionation was performed as described in Section 4, using NQO1 stable cells with or without bL (2 µM) for 24 h. The cell lysate was subjected to Western blot analysis with the indicated antibodies; β-tubulin and lamin B were used for cytoplasmic and nuclear markers, respectively. (**B**) Cellular localization of SIRT1 was examined in MDA-MB-231 and two independent clones of NQO1 stable cells (NQO1 #1 and #2) with or without bL (2 µM) for the indicated time. Images were taken under a new hybrid microscope (Echo, San Diego, CA, USA). The scale bar is 130 μm. (**C**) Cell morphologies of MDA-MB-231 and two independent clones of NQO1 stable cells (NQO1 #1 and #2) were examined after treatment with a combination of bL (2 µM), an NQO1 inhibitor (ES 936, 1 µM), and an SIRT1 inhibitor (EX 527, 1 µM) for 24 h, as indicated. Images were taken using a new hybrid microscope (Echo, Echo, San Diego, CA, USA). The scale bar is 230 μm.

**Figure 4 ijms-19-03813-f004:**
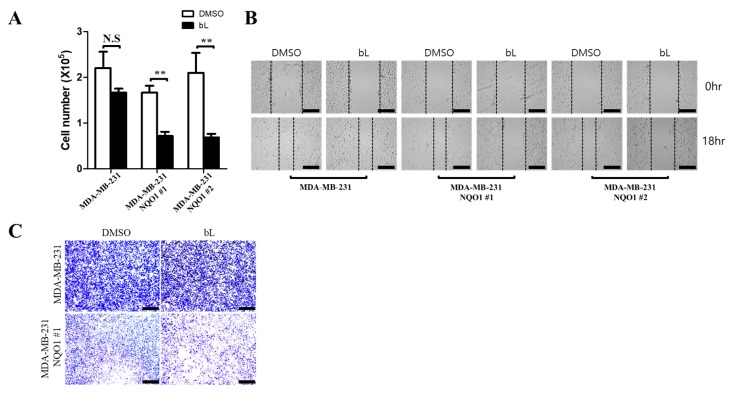
The bL compound suppresses cell proliferation and migration in an NQO1-dependent manner. (**A**) Cell proliferation was determined by counting the cell numbers of MDA-MB-231 and two independent clones of NQO1 stable cells (NQO1 #1 and #2) after treatment with bL (2 µM) for 24 h. The data are presented as means ± SEM; *n* = 3; ** *p* < 0.01, N.S: not significant. (**B**) A wound healing assay was performed as described in Section 4, using MDA-MB-231 and two independent clones of NQO1 stable cells (NQO1 #1 and #2) treated with bL (2 µM) for 18 h. Images were taken using a new hybrid microscope (Echo, San Diego, CA, USA). The scale bar is 230 μm. The black dash line indicates boundary that cells migrate. (**C**) A cell migration assay was performed with Transwell cell culture chambers as described in Section 4, using MDA-MB-231 and NQO1 stable cells treated with bL (2 µM) for 7 h. Images were taken using a new hybrid microscope (Echo, San Diego, CA, USA). The scale bar is 230 μm.

**Figure 5 ijms-19-03813-f005:**
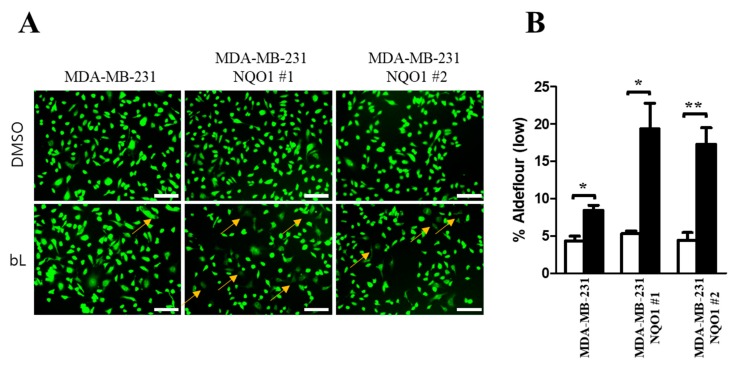
The bL compound decreases endogenous ALDH1 activity. (**A**) The overall activity of ALDH1 was examined using an ALDEFLUOR assay in MDA-MB-231 and two independent clones of NQO1 stable cells (NQO1 #1 and #2) treated with bL (2 µM) for 24 h. Green fluorescence was captured under a new hybrid microscope (Echo, San Diego, CA, USA). The scale bar is 130 μm. (**B**) For quantification, three independent areas were randomly taken, and cells with a lower intensity of ALDEFLUOR than the dimethyl sulfoxide (DMSO) control were counted. Quantitative analysis revealed that bL efficiently decreased the endogenous activity of ALDH1 in an NQO1-dependent manner. The yellow arrows indicate ALDEFLUOR (low) cells; * *p* < 0.05; ** *p* < 0.01.

**Figure 6 ijms-19-03813-f006:**
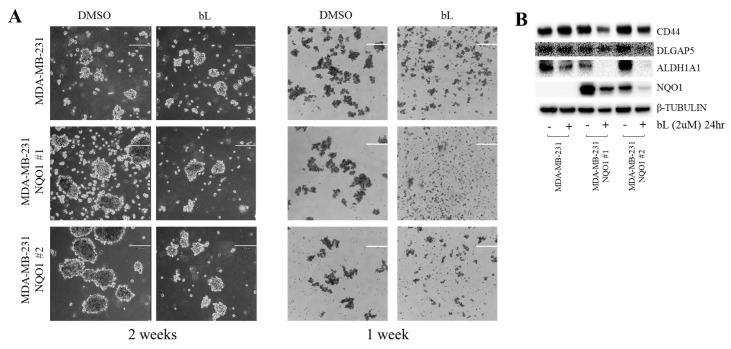
The bL compound inhibits mammosphere formation. (**A**) MDA-MB-231 and two independent clones of NQO1 stable cells (NQO1 #1 and #2) were induced to mammosphere formation for one and two weeks, as described in Section 4. After mammosphere formation, bL (2 µM) was added for 24 h. Images were taken using a new hybrid microscope (Echo, San Diego, CA, USA). The scale bar is 200 μm. (**B**) Cell lysates obtained from the mammosphere formation assay were subjected to Western blot analysis using the indicated antibodies; β-tubulin was used as a loading control.

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
