# Peer review of "NQO1 is Required for β-Lapachone-Mediated Downregulation of Breast-Cancer Stem-Cell Activity"

_ijms, 2018, doi:10.3390/ijms19123813_

Round 1

Reviewer 1 Report

This article is written well, but lacks in new knowledge and grounds. Therefore, I require Major Revision.
・Please clarify the purpose of this study in “Introduction” section.
・Please add experiments using clinical specimens.
・Please verify the view on tumor microenvironment additionally.
・This paper has not been analyzed and discussion of the problem.
・And, please describe the discussion in more detail.
・What does this research have for breast cancer clinical practice?

Minor point
・The sentence of this paper has many careful mention errors. Please review it.

Author Response

Response to Reviewer 1 Comments

We are delighted that the reviewer considered our findings interesting. We appreciate the reviewer’s constructive criticisms. Following are our point-by point responses to the reviewer’s comments.

Comments and Suggestions for Authors

This article is written well, but lacks in new knowledge and grounds. Therefore, I require Major Revision.

Please clarify the purpose of this study in “Introduction” section.

As you pointed out, we revised a sentence and added the purpose of this study in the introduction section (line 83-88, page 2). In the study, we focused on evaluating the effect of bL on the activity of BCSCs, rather than cancer cells itself because we tried to find new candidate drugs targeting CSCs that normally show therapy resistance to various cancer drugs. In figure 6, we found that bL efficiently inhibited mammosphere formation, which represents one of the CSCs’ properties. Even though well-designed in vivo experiments are required to see exact efficacy of bL, we think that in vitro experiments using cell lines are sufficient to evaluate the potential efficacy of bL on CSCs.

Please add experiments using clinical specimens.

We agree with your suggestion, but it takes a long time to take clinical specimens from patients with breast cancer, and thenisolate BCSCs from tissue or blood. Therefore, we do not think it is possible to perform these experiments within the given time (10days).

Please verify the view on tumor microenvironment additionally.

As you suggested, we described the possible effect of bL on tumor microenvironment in the discussion section (line 299-309, page 9).

This paper has not been analyzed and discussion of the problem.

We revised sentences and added additional paragraphs in discussion section and marked them in red.

And, please describe the discussion in more detail.

As you suggested, we tried to describe the discussion in more details and marked in red.

What does this research have for breast cancer clinical practice?

Although the development of anti-cancer drugs has dramatically improved, breast cancer still remains the cancer with the second leading mortality rate among women. This is mainly due to the high rate of metastatic property and recurrent rate of breast tumors. Accumulating studies suggested BCSCs mediate tumor metastasis and relapse because of drug resistance. So, in this study, we tried to evaluate bL as a new candidate drug to treat BCSCs using an in vitro culture system, mammosphere formation assay that is a suitable method to evaluate stem-cell-like features of breast cancer cells (Figure 6A). Even though extensive experiments, including in vivo experiments, are required to evaluate the exact effect of bL on the activity of BCSCs, we believe that our current study, using an in vitro cell system, is sufficient to measure potential efficacy of bL on the activity of BCSCs. Since clinical trials with bL have been done and some trials have been testing with cancers, including pancreatic cancer (https://ClinicalTrials.gov), the applicability of bL to cancers with high rate of metastasis and relapse such as breast and pancreatic cancer is promising.

Minor point

The sentence of this paper has many careful mention errors. Please review it.

As you suggested, we have carefully read it again and revised sentences and typos properly.

Reviewer 2 Report

1.      MCF-7 is an ER/PR positive breast cancer cell line, MDA-MB-231 is a TNBC cell line. The two cell lines are from different subtypes of breast cancer. I’m not convinced comparing two different subtype of breast cancer cells and making conclusions.

2.      Does the concentration of Beta-lapachone used in the in vitro experiments is clinically achievable? (authors should comment this based on half-life, maximum steady state plasma level concentration that could achieved and protein binding ability)

3.      Authors mentioned SIRT1 is not involved in bL-NQO1-mediated gene expression and cell death. But the sirtuins family consists of 7 members, did the authors check others before making conclusions that sirtuins are involved?

4.      Though this is not a major prerequisite of this paper, beta-lapachone was shown to induce ROS, and ROS plays a role in CSC. Further, NQO1 plays a role in generation of ROS. Did the authors investigate for the changes in ROS relation in this study?

Author Response

Response to Reviewer 2 Comments

We appreciate the reviewer’s insightful comments and valuable suggestions. Our point-by-point responses to the reviewer’s suggestions are as follows:

1.      MCF-7 is an ER/PR positive breast cancer cell line, MDA-MB-231 is a TNBC cell line. The two cell lines are from different subtypes of breast cancer. I’m not convinced comparing two different subtype of breast cancer cells and making conclusions.

As you pointed out, two cell lines have been established from different breast tissue origins . In this study, we tried to focus on finding a new drug for treating BCSCs that mediate tumor metastasis and relapse due to resistance to current chemotherapy. The two cell lines, MCF7 and MDA-MB-231, have shown non-tumorigenic (also BCSC negative) markers and tumorigenic (also BCSC positive) markers in nude mice, respectively, and MDA-MB-231 cells are TNBCs that are known to be resistant to chemotherapy (Sci Rep. 2017 Oct 23;7(1):13856). Furthermore, MDA-MB-231 cells do not have detectable endogenous NQO1 expression compared with MCF7 in our experiment (Fig 1E). Therefore, we thought that the MDA-MB-231 cell line is very suitable to examine the effect of bL after introducing NQO1,to see whether bL has a suppressive effect on cells with BCSC markers that represent BCSC properties. Ultimately, we wanted to use and compare two cancer cell lines representative of human breast cancer showing different cancer behaviors. MCF7 cells that were positive for NQO1 expression were also used as control cells to see whether there is a correlation in expression between NQO1 and BCSC markers.

2.      Does the concentration of Beta-lapachone used in the in vitro experiments is clinically achievable? (authors should comment this based on half-life, maximum steady state plasma level concentration that could achieved and protein binding ability)

A previous study with healthy Korean male subjects (Drug Des Devel Ther. 2017; 11: 2719–2725.) identified that the maximum plasma concentration (Cmax) of bL was 3.56±1.55 ng/mL and 14.96±15.56 ng/ml, and the half-life (t½ (h)) was 5.89±3.41ng/ml and 18.16±3.14ng/ml after a single dose (100mg) or multiple doses (100mg twice daily for 8 days), respectively. Our dose of bL used for breast cancer cells was approximately 484.54ng/ml when converted, showing a higher concentration than that examined in human subjects. Previously, many studies with various cancer cell lines such as Hep2 carcinoma cells (4µM, Cancer Res. 1987 Oct 15;47(20):5361-6), PC-3, DU145, and LNCaP human prostate cancer cells (4µM, Cancer Res. 2010 Oct 15;70(20):8088-96), HL-60 human promyelocytic leukemia cells (8µM, Cancer Res. 1995 Sep 1;55(17):3712-5), as well as HT29 and HCT116 human colon cancer cell lines (5µM, PLoS One. 2015 Feb 18;10(2):e0117051) used a higher concentration (>4µM) than we used (2µM) to see the effect of bL on each cell line. So, we think that the appropriate concentration of the drug should be determined according to the experimental purpose and type. In fact, we first determined the appropriate dose of bL for viability of MDA-MB-231 cells before starting experiments and determined that 2µM bL is an effective dose for our experimental purpose. Although the concentration used in this study is higher than that in plasma in the previous study with healthy subjects, we think 2µM bL would be a reference dose for treatment of patients with breast cancer in clinical trials in the future.

3.      Authors mentioned SIRT1 is not involved in bL-NQO1-mediated gene expression and cell death. But the sirtuins family consists of 7 members, did the authors check others before making conclusions that sirtuins are involved?

We have not tested with other sirtuin family members in our experiments. Actually, it is not likely that other sirtuins (SIRT2-7) were affected by EX-527 (1uM used in this study) because EX-527 is a very specific drug for SIRT1 (IC50 of 38 nM for SIRT1, and for other sirtuin family members >19.6uM). As you pointed out, we could not rule out the possible involvement of other sirtuin family members in bL-NQO1 induced cell death. However, to verify this point, we need specific drugs or siRNA for each sirtuin and to perform additional cell experiments. We think It is worthy of studying, but it could be done in the future study due to the time limit.

4.      Though this is not a major prerequisite of this paper, beta-lapachone was shown to induce ROS, and ROS plays a role in CSC. Further, NQO1 plays a role in generation of ROS. Did the authors investigate for the changes in ROS relation in this study?

Although we have not specifically performed experiments for this purpose, bL could generate ROS by activating NQO1 and it could thusly induce cell death according to previous studies as you pointed out. We discussed and mentioned the possible involvement of ROS by bL in cell death in the discussion and introduction sections, marked in red. Because CSCs have specific mechanisms to protect themselves from the genotoxic effects of reactive oxygen species (ROS) by overexpressing ROS scavenging genes such as superoxide dismutase, catalase, and glutathione peroxides (Contemp Oncol (Pozn), 2015;19(1A):A7-A15, Nature. 2009 Apr 9;458(7239):780-3) and normally have a low level of ROS, a strong activator of NQO1 such as bL will be needed to induce a tremendous endogenous ROS level and to escape from highly developed ROS scavenging systems in CSCs.

Round 2

Reviewer 1 Report

Please prove whether the definition of breast cancer stem cell is correct.

Author Response

Please prove whether the definition of breast cancer stem cell is correct.

Previous studies identified that the most common biomarkers used to identify the breast cancer stem cell (BCSC) phenotype are CD44+, CD24-, and ALDH1+ (Stem Cell Investig. 2017 Nov 29;4:96). In this study, we compared two cancer cell lines representative of human breast cancer, MCF7 (no BCSC, CD44+/CD24− low cells) and MDA-MB-231 (BCSC like, CD44+/CD24-/ALDH1+ High cells, (Sci Rep. 2016 Nov 23;6:37340, and Sci Rep. 2017 Oct 23;7(1):13856)) and want to see whether NQO1 is essential for bL mediated regulation of BCSCs activity using in vitro cell culture system.

Reviewer 2 Report

Comments were addressed

Author Response

Comments were addressed

We are glad that your concerns and questions were fully addressed.

We really appreciate your insightful comments.